# Chimera: State Space Models Beyond Sequences

**Aakash Lahoti**[*]                                        *alahoti@andrew.cmu.edu*
*Carnegie Mellon University*

**Tanya Marwah**[*]                                        *tmarwah@andrew.cmu.edu*
*Carnegie Mellon University*

**Ratish Puduppully**                                    *ratishpuduppully@gmail.com*
*IT University of Copenhagen*

**Albert Gu**                                             *agu@cs.cmu.edu*
*Carnegie Mellon University,*
*Cartesia AI*

**Reviewed on OpenReview:** *https://openreview.net/forum?id=yv0TUssepk*

## Abstract

Transformer-based deep learning methods have emerged as the standard approach to model diverse data such as sequences, images, and graphs. These methods rely on self-attention, which treats data as an unordered set of elements. This ignores the neighborhood structure or *graph topology* of the data and requires the use of inductive biases, such as position embeddings in sequences and images, and random walks in graphs, to incorporate topology. However, developing bespoke inductive biases for each task requires significant effort and can also introduce side-effects hindering generalization. In this work, we introduce *Chimera, a unified model that directly incorporates the data topology in a principled way*, obviating the need for domain-specific biases. Central to Chimera is the observation that state-space models—which naturally do not require position embeddings—can be generalized to capture any general graph topology. Our experiments demonstrate the versatility of our approach—Chimera achieves strong performance across the domains of language, vision, and graphs, outperforming BERT on GLUE by 0.7 points, ViT on ImageNet-1k by 2.6%, and all the baselines on the Long Range Graph Benchmark. Our results validate Chimera's *principled methodological contributions* and affirm the long-held belief that data topology is a powerful inductive bias across modalities. We further propose *algorithmic optimizations* to improve Chimera's efficiency while maintaining performance: 1) For the subclass of Directed Acyclic Graphs we show that Chimera can be implemented as a linear time recurrence. 2) For general graphs, we relax the method with a simple mathematical approximation, achieving Transformer's quadratic complexity without relying on domain-specific biases.

## 1 Introduction

Real-world data, ranging from sequential language and audio to high-dimensional images and structured molecule data, often exhibit some notion of neighborhood, or *graph topology*, among its constituent elements. For instance, language and audio have a directed line graph topology; images possess an undirected grid-graph topology; Structured molecule data has predefined nodes (atoms) and edges (bonds) that constitute its topology. More recently, Transformer-based (Vaswani et al., 2017) methods, with self-attention at their core, are increasingly being used to model such data (Devlin et al., 2019; Dosovitskiy et al., 2021; Rampášek et al., 2022). However, self-attention is permutation invariant and treats data as an unordered set of elements,

---

[*]Authors contributed equally to this work.

disregarding its topology. Consequently, significant research effort has focused on developing domain-specific inductive biases, such as position embeddings (Su et al., 2023; Devlin et al., 2019), and random walks (Behrouz and Hashemi, 2024; Wang et al., 2024), to incorporate data topology into the model.

However, designing these inductive biases requires navigating a large search space for each domain. For instance, RoPE embeddings (Su et al., 2023) work well in language (Touvron et al., 2023); in vision, absolute and 2D-RoPE embeddings are widely used (Dosovitskiy et al., 2021; Heo et al., 2024); while Laplacian embeddings or random walks are used in graphs (Rampášek et al., 2022). Moreover, these techniques can produce undesirable side effects, such as poor out-of-domain generalization—RoPE struggles to generalize to sequences longer than the training lengths (Kazemnejad et al., 2024), while absolute position embeddings have inherently constrained context sizes due to their design. Furthermore, it is unclear how effectively these techniques capture the underlying graph topology.

In this paper, we introduce *Chimera*, a unified framework that *directly* incorporates data topology—i.e., the underlying graph structure—in a principled way. Chimera is motivated by the observation that State Space Models (SSMs) for causal language modeling—Mamba-2 (Dao and Gu, 2024), RetNet (Sun et al., 2023), and Linear Attention (LA) (Katharopoulos et al., 2020)—naturally capture the sequence order through recurrence, without position embeddings. We formalize this property and generalize it beyond causal sequences to any graph topology. This approach is in contrast with existing methods that instead apply attention or SSMs as a black box to "flattened data" augmented with heuristics to compensate for loss of topological information (Liu et al., 2024; Dosovitskiy et al., 2021). We validate our *methodological innovation* with empirical results that demonstrate strong performance across diverse domains. This affirms the long-held belief that topology is a powerful inductive bias, while providing a principled way to incorporate it into the modeling process.

We first formally show how SSMs capture the underlying directed line graph topology of language data through recurrence (Sec 3). For this, we leverage the Structured Masked Attention (SMA) representation (Dao and Gu, 2024) of SSMs: methods such as Mamba-2, RetNet, and Linear Attention are equivalent to the matrix $\mathbf{M} = \mathbf{L} \odot (\mathbf{Q}\mathbf{K}^T)$ multiplied with the input, where $\mathbf{Q}, \mathbf{K}$ are the query and key matrices, respectively, and $\mathbf{L}$ is the (data-dependent) mask matrix analogous to the causal mask in attention. We show that the mask matrix fully encodes the topology of the underlying graph structure by acting as the *resolvent* of the adjacency matrix, $\mathbf{A}$, of a directed line graph, i.e. $\mathbf{L} = (\mathbf{I} - \mathbf{A})^{-1} = \sum \mathbf{A}^i$, where $\mathbf{I}$ is the identity matrix. This result allows us to generalize to any graph topology. Specifically, for an "appropriately parameterized" adjacency matrix, $\mathbf{A}$, of the graph topology, we compute the matrix multiplication of $\mathbf{M} = \mathbf{L} \odot (\mathbf{Q}\mathbf{K}^T)$, where $\mathbf{L} = (\mathbf{I} - \mathbf{A})^{-1}$, with the input. Intuitively, $\mathbf{A}_{ij}$ captures the "influence" between neighbors $i$ and $j$, while the resolvent aggregates this influence over all paths, thus capturing the underlying topology. In Section 3.3, we present the detailed parameterization scheme used in Chimera which is important for both empirical performance and numerical stability.

The main bottleneck lies in the computation of the mask matrix, whose naive implementation incurs a cubic cost. We propose two algorithmic optimizations to mitigate this cost while maintaining performance:

1. We specialize the method for the subclass of directed acyclic graphs (DAGs). This is motivated by the fact that many graph topologies can be canonically decomposed into multiple DAGs. For example, an undirected line graph can be decomposed into two directed line graphs (Fig 2), while a grid graph can be divided into four directed grid graphs (Fig 3). We prove that for this subclass, the resolvent can be computed by running a recurrence that is linear in the number of nodes and edges. We further propose a squaring technique to compute the resolvent efficiently on modern hardware accelerators by leveraging matrix multiplications, at a quadratic cost in the number of vertices. This cost is optimal in the worst case, it can be improved when the underlying graph is "structured"—line graphs can computed in linear time via existing Mamba-2 kernels

2. We relax the exact computation of the resolvent for general graphs with a finite sum approximation of the Neumann series, i.e. $(\mathbf{I} - \mathbf{A})^{-1} = \sum_{i=0}^{d} \mathbf{A}^i$, where $d$ is the diameter of the graph. We can efficiently compute this approximation with a squaring technique, capturing the global topological structure. We further show that the finite sum approximation performs as well as the method with the sum of infinte terms.

Overall, we make the following contributions:

- We propose Chimera, a unified model that directly incorporates graph topology in a principled way by generalizing SSMs. This is in contrast with existing approaches that apply attention or SSMs as a black box on "flattened data" with additional heuristics.
- We introduce algorithmic optimizations to improve the method's efficiency by specializing it to DAGs and approximating the resolvent using a finite sum, while preserving performance. We show that for canonical modalities such as images and language, where data can be decomposed into DAGs, this finite-sum approximation becomes exact.
- We demonstrate that Chimera consistently achieves strong performance across diverse domains including language, images, and graphs—outperforming BERT (Devlin et al., 2019) with a GLUE score (Wang et al., 2019) of 0.7, surpasses ViT (Dosovitskiy et al., 2021) on ImageNet-1k (Deng et al., 2009) classification by 2.6%. Furthermore, our method outperforms strong baselines on the Long Range Graph Benchmark (LRGB) (Dwivedi et al., 2022), where we show that our model is capable of modeling both long and short range interactions nodes, while respecting the graph structure.

## 2 Preliminaries

We introduce State Space Models (SSMs), which are recurrent models designed to process sequential data, such as language and audio. We formulate SSMs in their recurrent form and then introduce the Structured Masked Attention (SMA) (Dao and Gu, 2024) representation that unrolls and vectorizes this recurrence as a matrix $\mathbf{M}$ acting on the input $\mathbf{X}$. This SMA representation would allow us to show that SSMs inherently operate on a directed line graph topology.

### 2.1 Overview of State Space Models

SSMs, such as Mamba-2 (Dao and Gu, 2024), Linear Attention (LA) (Katharopoulos et al., 2020), RetNet (Sun et al., 2023), are recurrent sequence-to-sequence models that feature a linear hidden-state transition function. This function is typically data-dependent which is known to improve model performance (Hwang et al., 2024).

Formally, let $\boldsymbol{X} \in \mathbb{R}^{T \times D}$ denote the input sequence of $T$ tokens, where each token has $D$ channels. Let the size of the hidden state be $d$. Let $\boldsymbol{Y} \in \mathbb{R}^{T \times D}$ be the output of the sequence-to-sequence model. Then, SSMs first compute the following matrices:

$$\boldsymbol{B} = f_B(\boldsymbol{X}), \boldsymbol{C} = f_C(\boldsymbol{X}), \boldsymbol{V} = f_V(\boldsymbol{X}) \in \mathbb{R}^{T \times d}, \tag{1}$$

where $f_B$, $f_C$, $f_V$ are model specific data dependent functions. For instance, in Mamba-2 each of these functions is a composition of a linear projection of $\boldsymbol{X}$ along the channel dimension, followed by a short convolution layer along the sequence dimension and a Swish activation function (Ramachandran et al., 2017). In Dao and Gu (2024), it was shown that we can view the $\boldsymbol{B}$, $\boldsymbol{C}$, $\boldsymbol{V}$ matrices as analogs of the key, query, value matrices in self-attention, respectively.

Let $\boldsymbol{v}^i = \boldsymbol{V}[:,i] \in \mathbb{R}^T$ denote the input corresponding to channel $i$. Let $\boldsymbol{B}_t = \boldsymbol{B}[t,:]$, $\boldsymbol{C}_t = \boldsymbol{C}[t,:]$ for any time $t$. Let $y_t^i = \boldsymbol{Y}[t,i]$ and $v_t^i = \boldsymbol{v}^i[t]$. Then, the model computes the following recurrence, starting with the hidden state vector $\boldsymbol{h}_{-1}^i = \boldsymbol{0} \in \mathbb{R}^d$:

$$\boldsymbol{h}_t^i = a_t \boldsymbol{h}_{t-1}^i + b_t \boldsymbol{B}_t v_t^i, \tag{2}$$
$$y_t^i = \boldsymbol{C}_t^T \boldsymbol{h}_t^i, \tag{3}$$

where $a_t, b_t$ are model-specific parameters that characterize the SSM. LA sets $a_t = b_t = 1$, RetNet sets $a_t = \gamma$, $b_t = 1$ for a learnable parameter $\gamma$. In contrast, Mamba-2 sets $a_t, b_t$ in a data-dependent manner that implicitly encodes a gated memory mechanism known as *selectivity* or the *selection mechanism*. This allows the model to select and propagate important tokens across long sequences. Specifically,

$$\Delta = f_\Delta(\boldsymbol{X}) \in \mathbb{R}^T; a_t = \exp(-\Delta_t), b_t = \Delta_t \in \mathbb{R}, \tag{4}$$

where $\Delta$ is the selectivity matrix, $\Delta_t$ is the $t^{\text{th}}$ element of the vector, and $f_\Delta$ like $f_B$, $f_C$, $f_V$ is a data-dependent function. Selectivity works by assigning larger values $\Delta_t$ to important tokens, amplifying their

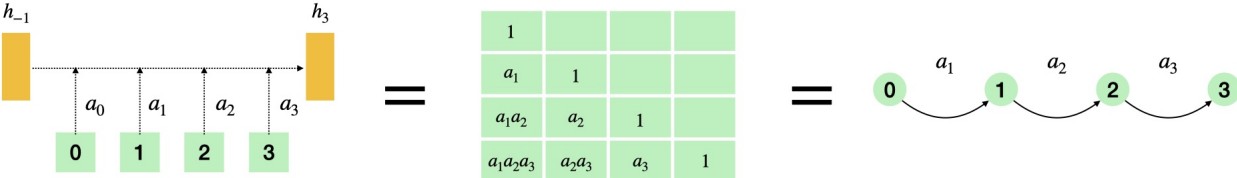

Figure 1: *SSMs inherently operate on a directed line graph topology*: SSMs modeling a sequence of tokens in the recurrent representation (left), the structured mask matrix from the SMA representation of SSMs (center), The underlying directed line graph topology (right).

contribution to the previous hidden state, while assigning smaller values $\Delta_t$ to unimportant tokens, which preserve the past hidden state with minimal influence from these tokens.

## 2.2 The Structured Masked Attention Representation

Dao and Gu (2024) introduced the Structured Masked Attention (SMA) representation for SSMs, which vectorizes the time-stepped recurrence (Eq. 3) as a matrix multiplication, $\boldsymbol{Y} = \boldsymbol{MV}$. [1] Here, $\boldsymbol{M}$ depends on the data-driven matrices $\boldsymbol{B}, \boldsymbol{C}$, and $\Delta$, and can be expressed as $\boldsymbol{M} = \boldsymbol{L} \circ (\boldsymbol{CB}^T)$, where $\boldsymbol{L}$ is a mask derived from $\Delta$. One can obtain this formulation by unrolling the recurrence across all time steps.

Formally, define $\bar{\boldsymbol{B}}_t = b_t \boldsymbol{B}_t$, and recall from Section 2.1 that $b_t = \Delta_t$, $a_t = \exp(-\Delta_t)$ for Mamba-2; $b_t = 1$, $a_t = \gamma$ for RetNet; and $b_t = 1$, $a_t = 1$ for Linear Attention. Then the output $\boldsymbol{Y}$ of the recurrence (Eq. 3) can be vectorized as,

$$\boldsymbol{Y} = \boldsymbol{MV} = (\boldsymbol{L} \odot \boldsymbol{C\bar{B}}^T)\boldsymbol{V}, \tag{5}$$

where the mask matrix $\boldsymbol{L}_{ij} = \mathbf{1}[i \geq j]\, \Pi_{j < k \leq i} a_k$, The SMA representation of the recurrence (Eq. 3) is useful because it neatly isolates the effect of the underlying topology within the recurrence computation into the mask matrix $\boldsymbol{L}$ (Sec. 3) This property will allow us to generalize SSMs to arbitrary topologies by appropriately formulating the structured mask $\boldsymbol{L}$.

## 3 Chimera: Incorporating Graph Topology

In this section, we introduce Chimera, a unified model that directly incorporates the underlying graph topology of a domain by mathematically generalizing SSMs. This contrasts to existing methods, such as Behrouz and Hashemi (2024); Devlin et al. (2019); Liu et al. (2021), that use attention or SSMs as a black-box applied to 'flattened data' and rely on inductive biases to incorporate structural information.

Our motivation stems from the fact that SSMs on causal language modeling task do not require position embeddings and naturally capture the sequence order with their recurrence. We seek to formalize this result which then allows us to generalize it to arbitrary graphs. To this end, we begin by defining the resolvent of a linear operator and interpret its action when this operator is the adjacency matrix.

### 3.1 Resolvent of an Adjacency Matrix Accumulates Influence Along All Paths

A graph consists of a set of nodes $\mathcal{V}$ that represent data elements, and edges $\mathcal{E}$ that encode the underlying topological structure. We conceptualize the associated adjacency matrix $\boldsymbol{A} \in \mathbb{R}^{|\mathcal{V}| \times |\mathcal{V}|}$ as *capturing the influence between neighboring nodes*. Specifically, $\boldsymbol{A}_{ij}$ is the influence that node $j$ has on node $i$, for each edge $(i, j)$. The desideratum is to extend the notion of influence to all node pairs by incorporating the graph's structure, accounting for all possible paths between them. To model this cumulative influence, we introduce the concept of the resolvent of a linear operator

---

[1]Not all SSMs admit an SMA representation. We focus on those that do, such as LA, RetNet, and Mamba-2. In this work, we use the term "SSMs" specifically to refer to this restricted class.

**Definition 3.1** (Resolvent of a Linear Operator (Reed and Simon, 1980))**.** *Let $\boldsymbol{A} \in \mathbb{R}^{T \times T}$ be a linear operator, $\boldsymbol{I}$ the identity operator, and $\lambda$ a complex number. Then, the resolvent operator is defined as:*

$$R(\lambda, \boldsymbol{A}) = (\lambda \boldsymbol{I} - \boldsymbol{A})^{-1}, \tag{6}$$

*which exists for all complex numbers $\lambda$ that are not in the spectrum of $\boldsymbol{A}$, i.e., $\lambda \notin \sigma(\boldsymbol{A})$. In this work, we set $\lambda = 1$ to remain in the field of real numbers, and this is done without loss of generality, as any choice of $\lambda$ is equivalent upto scaling of the model.*

We now demonstrate how the resolvent operator captures the influence between any two nodes in the graph. Observe that the resolvent operation can be expanded using the Liouville-Neumann series if the operator norm of the adjacency matrix is strictly less than 1, i.e. $\|\boldsymbol{A}\| < 1$,

$$R(1, \boldsymbol{A}) = (\boldsymbol{I} - \boldsymbol{A})^{-1} = \sum_{k=0}^{\infty} \boldsymbol{A}^k. \tag{7}$$

Intuitively any term, $\boldsymbol{A}_{ij}^k$, in this expansion represents the influence between nodes $i$ and $j$ accumulated across all paths of length exactly $k$ connecting them. We formalize this in the following Proposition 3.2.

**Proposition 3.2** ($\boldsymbol{A}^k$ accumulate influence through paths of length $k$)**.** *Given the weighted adjacency matrix $\boldsymbol{A} \in \mathbb{R}^{T \times T}$ of a graph $\mathcal{G} = (\mathcal{V}, \mathcal{E})$ with $|\mathcal{V}| = T$, where $A_{ij}$ is the weight of the $(i, j)$ edge of the graph. Then $(i, j)^{th}$ entry of $\boldsymbol{A}^k$ is given as,*

$$(\boldsymbol{A}^k)_{ij} = \sum_{p_1, p_2, \dots, p_{k-1}} \boldsymbol{A}_{ip_1} \boldsymbol{A}_{p_1 p_2} \cdots \boldsymbol{A}_{p_{k-1} j},$$

*where $(p_1, \dots, p_{k-1})$ is an ordered sequence of vertices forming a path of length $k$ from node $i$ to $j$.*

Therefore, the series $(\boldsymbol{I} - \boldsymbol{A})_{ij}^{-1}$ (Eq. 7) sums up the influence of node $i$ on node $j$ over all possible lengths.

### 3.2 SSMs operate on a Directed Line Graph

We now show that SSMs naturally operate on a directed line graph. Specifically, let $\mathcal{V}$ be the set of tokens, and $\mathcal{E}$ be the edges connecting token $t$ to the next token $t + 1$. Let the weighted adjacency matrix be $\boldsymbol{A}_{s,t} = \mathbf{1}_{[t=s+1]} a_t$, where $a_t$ is the SSM-specific parameter defined in Section 2.2.

Recall from Section 2.2 that SSMs are equivalent to the matrix action of $\mathbf{M} = \mathbf{L} \odot (\mathbf{CB}^T)$ on the input. We make the key observation that $\boldsymbol{L}$ *is precisely the resolvent of* $\boldsymbol{A}$, that is $\boldsymbol{L} = (\boldsymbol{I} - \boldsymbol{A})^{-1}$. This mathematically ties SSMs' recurrence to the directed line graph topology, with the mask encoding the topology (Fig 1).

**Proposition 3.3.** *Under the notation established in Section 2, consider a weighted directed graph $\mathcal{G}$ with nodes $\mathcal{V} = \{0, \cdots, T-1\}$, edges $\mathcal{E} = \{(i-1, i) | i \in \mathcal{V}, i > 0\}$, and the edge weights $\mathcal{W} = \{w_{(i-1,i)} = a_i | i \in \mathcal{V}, i > 0\}$. Let $\boldsymbol{A}$ be the weighted adjacency matrix of incoming edges,*

$$\boldsymbol{A} = \begin{bmatrix} 0 & 0 & 0 & \cdots & 0 \\ a_1 & 0 & 0 & \cdots & 0 \\ 0 & a_2 & 0 & \cdots & 0 \\ \vdots & \vdots & \vdots & \ddots & \vdots \\ 0 \cdots 0 & 0 \cdots 0 & 0 & a_{T-1} & 0 \end{bmatrix}, \tag{8}$$

*then $\boldsymbol{L} = \sum_{i=0}^{\infty} \boldsymbol{A}^i = (\boldsymbol{I} - \boldsymbol{A})^{-1}$, and consequently, $\boldsymbol{y} = ((\boldsymbol{I} - \boldsymbol{A})^{-1} \odot \boldsymbol{C}\bar{\boldsymbol{B}}^T)\boldsymbol{V}$.*

We interpret this result intuitively: In a directed line graph, there is exactly one path between the tokens $i, j$ with $i < j$, and the corresponding mask matrix entry $\boldsymbol{L}_{ij} = \prod_{i \geq k > j} a_k$, reflects the cumulative influence of the intervening tokens along this path. Furthermore, $\boldsymbol{L}_{ij} = 0$ for $i < j$ restricts influence in the forward direction, ensuring causality. This shows that SSMs inherently operate on a directed line graph with the $\boldsymbol{L}$ matrix encoding the topology.

### 3.3 Generalizing SSMs to Arbitrary Graph Topologies

Building on Proposition 3.3, we can generalize SSMs from causal sequences to arbitrary graph topologies. Specifically, we compute the resolvent of the adjacency matrix, $\boldsymbol{A}$, with output $((\boldsymbol{I} - \boldsymbol{A})^{-1} \odot (\boldsymbol{C}\bar{\boldsymbol{B}}^T))\boldsymbol{V}$.

We focus on the parameterization of $\boldsymbol{A}$ with the following key points: 1. It ensures the numerical stability of the method by addressing cases of non-invertibility or poor conditioning of resolvent; 2. It generalizes Mamba-2's selectivity that allows for modeling long-range dependencies.[2]

Formally, consider a graph $\mathcal{G} = (\mathcal{V}, \mathcal{E})$ with $|\mathcal{V}| = T$ nodes, where each node has $D$ channels. Let $d$ denote the generalized hidden state size. For each node, we compute,

$$\boldsymbol{B} = f_B(\boldsymbol{X}), \boldsymbol{C} = f_C(\boldsymbol{X}), \boldsymbol{V} = f_V(\boldsymbol{X}) \in \mathbb{R}^{T \times d}, \tag{9}$$

$$\Delta = f_\Delta(\boldsymbol{X}) \in \mathbb{R}^T, \tag{10}$$

where the functions $f_B$, $f_C$, $f_V(\boldsymbol{X})$, $f_\Delta$ are linear projections applied to the input, followed by a local graph convolution over neighboring nodes and a Swish activation as chosen in Mamba-2. Furthermore, if the data set features edge embeddings $\mathbf{E} \in \mathbb{R}^{|\mathcal{E}| \times D}$, we define $\Delta' = f_{\Delta'}(\mathbf{Z}) \in \mathbb{R}^{|\mathcal{E}|}$, where $\mathbf{Z} \in \mathbb{R}^D$ is an edge embedding, as the selectivity matrix corresponding to the edges. Here $f_{\Delta'}$ is computed similarly to $f_\Delta$ as in equation 10.

We parameterize the $\boldsymbol{A}$ matrix for each edge $(i, j) \in \mathcal{E}$ as,

$$\boldsymbol{A}_{ij} = \exp\left(-\frac{\Delta_i + \Delta_j + \Delta'_{(i,j)}}{3}\right), \tag{11}$$

to incorporate context from both ends of the edge $(i, j)$ as well as the edge embeddings. Here $\Delta_i$, $\Delta_j$ and $\Delta'_{(i,j)}$ are are the learned selectivity parameters for the nodes $i$, $j$ and edge $(i, j)$ respectively. To add directionality to $A_{ij}$ and to further increase the representational power of our model, we can also maintain two (different) $\Delta$'s such that $\boldsymbol{A}_{ij} = \exp\left(-\frac{\Delta_i^{(1)} + \Delta_j^{(2)} + \Delta'_{(i,j)}}{3}\right)$.

Note that the matrix $\boldsymbol{I} - \boldsymbol{A}$ may be non-invertible or poorly conditioned, which would inhibit inverse computation and stable training of the model. We mitigate this issue with a data-dependent normalization parameter $\Psi = f_\Psi(\boldsymbol{X}) \in \mathbb{R}^T$, computed similarly to $\Delta$, and perform a row-wise normalization of the adjacency matrix using $\Psi$. Specifically, for each row $i \in [T]$, we apply:

$$\boldsymbol{A}[i, :] = \frac{\gamma \boldsymbol{A}[i, :]}{\mathbf{1}^T \boldsymbol{A}[i, :] + \exp(-\Psi_i)},$$

where $\gamma$ is a scaling hyperparameter. The following proposition shows that this normalization guarantees the convergence of the Neumann series for the adjacency matrix $\boldsymbol{A}$.

**Proposition 3.4.** *Under Gaussian initialization, the row-wise normalization strategy ensures that $\|\boldsymbol{A}\| < 1$ and $\|(\boldsymbol{I} - \boldsymbol{A})^{-1}\|$ is bounded with probability $> 1 - \Phi(\frac{-1}{2\gamma})$.*

The proof for this proposition in Appendix A.1. Finally, we compute the resolvent matrix $\boldsymbol{L} = (\boldsymbol{I} - \boldsymbol{A})^{-1}$ and the output $\boldsymbol{y}$ as $(\boldsymbol{L} \odot \boldsymbol{C}\bar{\boldsymbol{B}}^T)\boldsymbol{V}$.

## 4 Chimera With Improved Efficiency

While Chimera supports arbitrary graph topologies, computing the resolvent incurs a cubic cost in the number of nodes, which can be prohibitively expensive for large graphs.

In this section, we propose two algorithmic optimizations to mitigate this cost: First, we specialize Chimera to a tractable yet expressive subclass of Directed Acyclic Graphs (DAGs) for which the resolvent can be computed in linear time by running a recurrence on the topologically sorted graph. Second, for general graphs, we relax the resolvent computation using a finite approximation, achieving quadratic complexity of Transformers without domain-specific heuristics.

---

[2]Our approach applies to any SSM with an SMA representation and in this work, we specifically use Mamba-2.

### 4.1 Chimera on DAGs

In this section, we introduce a tailored normalization scheme as well as a linear-time recurrent algorithm for Chimera on DAGs. We further propose a modern accelerator-friendly technique to compute this resolvent efficiently by leveraging matrix multiplications, although at the cost of quadratic FLOPs.

Our choice of the DAG subclass is motivated by its expressivity. Topologies such as undirected line and grid graphs can be canonically decomposed into DAGs: line graph divides into two directed line graphs (Fig 2) and grid graph divides into four directed grid graphs (Fig 3). This allows for efficient Chimera that preserves the grid topology of an image.

#### 4.1.1 Chimera on DAGs: The method

Formally, consider a DAG, $\mathcal{G} = (\mathcal{V}, \mathcal{E})$, with $|\mathcal{V}| = T$ nodes, each with $D$ channels and a hidden state size of $d$. For any node $i$, let $p(i)$ be the set of its parents. Let $\boldsymbol{B}, \boldsymbol{C}, \boldsymbol{V}, \Delta$ be the input projections as defined in Section 3. We define the adjacency matrix $\boldsymbol{A}$ as $\boldsymbol{A}_{ij} = \exp(-\Delta_{ij})$ for each $(i, j) \in \mathcal{E}$, and set $\bar{\boldsymbol{B}}_i = (\sum_{(i,j)\in\mathcal{E}} \Delta_{ij})\boldsymbol{B}_i$ for each node $i$. In this work, we define $\Delta_{ij} = (\Delta_i + \Delta_j)/2$, but more generally one can take $\Delta_{ij} = f(\Delta_i, \Delta_j)$ for any suitable (e.g. symmetric) function $f$ of the nodewise parameters. Then the output $\boldsymbol{y} = (\boldsymbol{L} \odot (\boldsymbol{C}\bar{\boldsymbol{B}}^T))\boldsymbol{V}$.

We first show that the resolvent $(\boldsymbol{I} - \boldsymbol{A})^{-1}$ exists.

**Proposition 4.1.** *For a DAG, $\boldsymbol{A}$ is nilpotent, that is $\boldsymbol{A}^K = \boldsymbol{0}$. Therefore, the inverse $(\boldsymbol{I} - \boldsymbol{A})^{-1}$ exists and is given by the finite sum:*

$$\boldsymbol{L} = (\boldsymbol{I} - \boldsymbol{A})^{-1} = \sum_{k=0}^{K-1} \boldsymbol{A}^k. \tag{12}$$

While the resolvent always exists, we note that its entries can become exceedingly large which can cause numerical instabilities. To show this, we first represent this method in its recurrent view in Prop. 4.2 (and is visualized in Figure 4).

**Proposition 4.2.** *Our method computes the following recurrence on each channel $v_i$ of $\boldsymbol{V}$:*

$$\boldsymbol{h}_i = \sum_{j\in p(i)} \boldsymbol{A}_{ij}\boldsymbol{h}_j - \bar{\boldsymbol{B}}_i v_i, \qquad y_i = \boldsymbol{C}_i^T \boldsymbol{h}_i, \tag{13}$$

*where $\boldsymbol{h}_l = \boldsymbol{0}$ for all leaf nodes $l$.*

Recall from Section 3.1 that each $\boldsymbol{L}_{ij}$ represents the cumulative sum of all paths from node $j$ to $i$, and in the worst case, the number of such paths and its resolvent entry grows exponentially with distance. To address this, we introduce a normalization scheme built directly into the recurrence:

**Proposition 4.3.** *The normalized method is:*

$$\boldsymbol{h}_i = \frac{1}{\sqrt{|p(i)|}} \sum_{j\in p(i)} \left(\boldsymbol{A}_{ij}\boldsymbol{h}_j - \ln(\boldsymbol{A}_{ij})\boldsymbol{B}_i v_i\right), \tag{14}$$

$$y_i = \boldsymbol{C}_i^T \boldsymbol{h}_i. \tag{15}$$

*This normalization ensures that $Var(\boldsymbol{C}_i^T \boldsymbol{h}_i) \leq 1$ under the assumption that the vectors $\{\boldsymbol{B}_i v_i, \boldsymbol{C}_i\}_i$ are i.i.d. Gaussians, that is $\boldsymbol{B}_i v_i, \boldsymbol{C}_i \sim \mathcal{N}(\boldsymbol{0}, \boldsymbol{I}_d)$.*

Figure 2: Canonical DAG decomposition of undirected line graph topology (left) into two directed line graph topologies (right).

The assumption that for all $i$ we have $\boldsymbol{B}_i, v_i, \boldsymbol{C}_i \sim \mathcal{N}(\boldsymbol{0}, \boldsymbol{I}_d)$ is justified given that weights are usually initialized with zero-mean and scaled-identity covariance (Xavier initialization Glorot and Bengio (2010)), in addition to the fact that such distributions are approximately preserved throughout training via normalization techniques. The proof follows by induction on the time step $t$, ensuring that the output variance is bounded by 1, $\text{Var}(\boldsymbol{C}_i^T \boldsymbol{h}_i) \leq 1$. The detailed proof in Appendix A.2. To incorporate this normalization in the SMA representation, we define,

$$\bar{\boldsymbol{A}} = \frac{1}{\sqrt{|p(i)|}} \boldsymbol{A}, \quad \bar{\boldsymbol{B}} = \frac{\ln(\boldsymbol{A}_{ij})}{\sqrt{|p(i)|}} \boldsymbol{B}, \quad \boldsymbol{L} = (\boldsymbol{I} - \bar{\boldsymbol{A}})^{-1}, \tag{16}$$

and compute the output $\boldsymbol{y} = (\boldsymbol{L} \odot (\boldsymbol{C}\bar{\boldsymbol{B}}^T))\boldsymbol{V}$.

### 4.1.2 Chimera is efficient on DAGs

Finally, we highlight that DAGs are a particularly important case of Chimera because of additional efficiency benefits, both through recurrent and vectorized implementations.

**Linear-Time Complexity in the Recurrent View** The intuition for linear complexity is that the resolvent operation for DAGs is *finite* because of the lack of cycles. From the adjacency matrix perspective, $\boldsymbol{A}$ is nilpotent, i.e. $\boldsymbol{A}^k = 0$, where $k$ is the diameter of the graph (Prop 4.1). Thus, when running Chimera as a recurrence on the DAG, the resolvent operation converges after one pass over the graph in the topologically-sorted order, which takes linear time.

**Proposition 4.4.** *The Chimera structured mask matrix L can be computed in $O(|\mathcal{V} + |\mathcal{E}|)$ complexity where $|\mathcal{V}|, |\mathcal{E}|$ is the number of vertices and edges of the graph, respectively.*

The proof is provided in Appendix A.3. We note that the linear-time complexity of Mamba-2 can be seen as a special case of Proposition 4.4 specialized to the directed line graph, where both $|\mathcal{V}|$ and $|\mathcal{E}|$ is equal to the sequence length.

**Improving Efficiency Through Matrix Multiplications** Finally, we note that on modern hardware accelerators such as GPUs and TPUs, various computational algorithms can have different efficiency tradeoffs. For example, on directed line graphs, the naive computation of SSMs and RNNs as a recurrence is not parallelizable and is inefficient in practice (Gu and Dao, 2023). In the case of DAGs, we present a technique to reduce both the forward and backward pass for Chimera to leverage only matrix multiplications which are heavily optimized on modern accelerators. Although this technique is highly parallelizable, it requires the materialization of the adjacency matrix which is quadratic in the number of nodes, $|\mathcal{V}|$. In the worst-case, i.e. a complete transitive DAG with $|E| = \frac{|\mathcal{V}|(|\mathcal{V}|-1)}{2}$), this bound is tight. However, for specialized structured subclasses (e.g., line graphs) where the number of edges is $O(\mathcal{V})$, this cost can be significantly reduced (e.g. via existing Mamba-2 kernels for line graphs). The proof for the following theorem is in Appendix A.4.

**Theorem 4.5.** *In case of Chimera on DAGs, the forward pass can be computed with $O(\log(dia(\mathcal{G})))$ matrix multiplications where $dia(\mathcal{G})$ is the diameter of the graph (i.e. length of the longest path), and the backward pass can be computed with $O(1)$ matrix multiplications.*

## 4.2 Approximate Chimera for General Topology

While DAGs allow for efficient computation in structured domains like images and language, directly computing the resolvent $\boldsymbol{L}$ for general graph topology remains computationally expensive. To address this, we use a

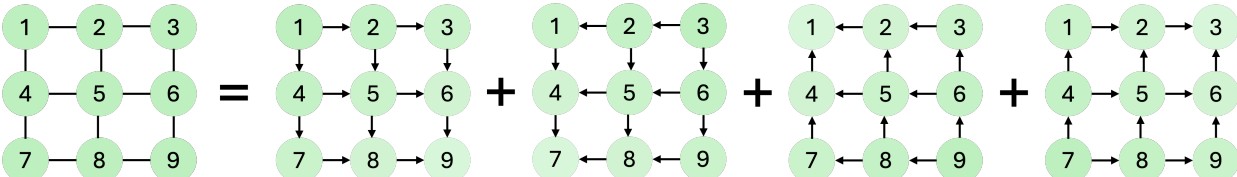

Figure 3: Grid graph (left). The canonical 2D-DAG decomposition of the grid graph (right). These graphs are sufficient to capture the influence between all pairs of nodes in the undirected grid graph.

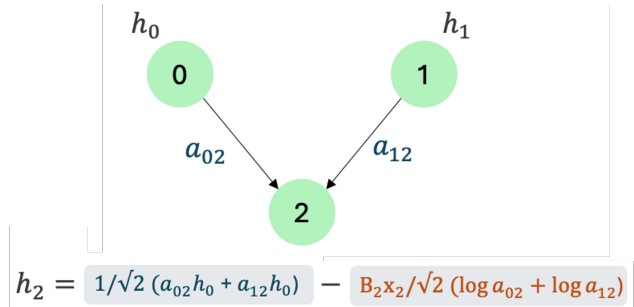

Figure 4: Chimera on DAGs: A visualization of the normalized recurrence. The absence of cycles in DAGs enables a recurrent view of the method which allows for a fast linear-time computation.

Table 1: Comparing Chimera on the undirected line graph (UG), and on DAG decomposed directed line graphs (DAG) with other state-of-the-art models including M2 (Fu et al., 2023), MLP-Mixer (Tolstikhin et al., 2021), FNet (Lee-Thorp et al., 2022), BERT (Devlin et al., 2019) on GLUE benchmark. Chimera outperforms all baselines including BERT with a linear time complexity. Here the best numbers are highlighted in **bold** and the second best numbers for each task are underlined.

| Method | #Params | Pretrain | | GLUE Tasks | | | | | | | | GLUE |
| | | $\mathcal{L}_{ce}$ | Acc (%) | MNLI | QNLI | QQP | RTE | SST2 | MRPC | COLA | STS | Avg |
|---|---|---|---|---|---|---|---|---|---|---|---|---|
| BERT-Base | 110M | 1.59 | 67.3 | 84.1 | **89.8** | **91.2** | 77.2 | 91.2 | 87.5 | 54.6 | **88.9** | 83.2 |
| MLP-Mixer | 112M | 1.77 | 63.5 | 77.2 | 82.4 | 87.6 | 67.3 | 90.5 | 86.5 | 43.0 | 85.2 | 77.5 |
| FNet | 112M | 1.94 | 61.3 | 74.9 | 82.1 | 85.7 | 63.6 | 87.6 | 86.4 | 42.7 | 83.1 | 75.8 |
| M2 | 116M | 1.65 | 65.9 | 80.5 | 86.0 | 87.0 | 69.3 | 92.3 | 89.2 | 56.0 | 86.9 | 80.9 |
| Chimera (UG) | 110M | 1.49 | 68.5 | 83.63 | 88.98 | 89.32 | 73 | 93.67 | 89.4 | 56.95 | 88.82 | 82.97 |
| Chimera (DAG) | 110M | 1.46 | **68.9** | **84.11** | 89.78 | 89.77 | **77.98** | **93.69** | **90.36** | **57.08** | 88.68 | **83.93** |

finite-sum relaxation of the resolvent operator and truncate its corresponding Neumann series sum (Eq. 7) at some maximum power $k \in \mathbb{N} > 0$. Specifically, let $\boldsymbol{A}$ be the (weighted) adjacency matrix of the graph topology defined in Section 3.3, then,

$$\boldsymbol{L} = \sum_{i=0}^{\infty} \boldsymbol{A}^i \approx \hat{\boldsymbol{L}} = \sum_{i=0}^{k} \boldsymbol{A}^i. \tag{17}$$

We choose $k = \mathrm{dia}(\mathcal{G})$, the diameter of the graph, to ensure that $\hat{\boldsymbol{L}}$ has access to the global structure of the graph, that is, it includes contributions from every edge and node in the graph.

**Proposition 4.6.** *If $k \geq dia(\mathcal{G})$, then for any pair of nodes $(i, j)$, if $\boldsymbol{L}_{ij} > 0$ in the original method, then $\hat{\boldsymbol{L}}_{ij} > 0$ in the finite-sum relaxation.*

As in Section 4.1.2, we can compute this approximation efficiently using the squaring trick:

$$\hat{\boldsymbol{L}} = (\boldsymbol{I} + \boldsymbol{A})(\boldsymbol{I} + \boldsymbol{A}^2)(\boldsymbol{I} + \boldsymbol{A}^4) \cdots (\boldsymbol{I} + \boldsymbol{A}^p), \tag{18}$$

where $p$ is the smallest power of 2 larger than or equal to the graph diameter $\mathrm{dia}(\mathcal{G})$. This reduces the computational cost of the method to $O(\log(\mathrm{dia}(\mathcal{G})))$ matrix multiplications.

For general graphs, since we cannot exploit the underlying structure of adjacency matrix $A$ to design efficient algorithms, the worst-case complexity of the finite sum approximation remains quadratic. This highlights a crucial point: as the complexity of the underlying structure increases, the computational cost of calculating the matrix $L$ will also grow accordingly.

## 5 Experiments

In this section, we will demonstrate that *directly incorporating topology is a powerful inductive bias for diverse domains* such as language, images and graphs, eliminating the need for domain-specific heuristics. Chimera consistently achieves state-of-the-art performance in these domains. On language, it outperforms BERT on the GLUE benchmark (Wang et al., 2019) by a GLUE score of 0.7. On images, it surpasses ViT models on the ImageNet-1k classification (Deng et al., 2009) task by 2.6%. On general graphs, Chimera outperforms strong baselines on the Long Range Graph Benchmark (Dwivedi et al., 2021) which highlights our method's ability to model long range interactions on graphs.

### 5.1 Masked Language Modeling

We evaluate Chimera on bidirectional language modeling, which has a line graph topology (Fig. 2). We test two Chimera variants: the general method[3] (Sec. 3) applied to an undirected line graph, and the DAG method (Sec. 4.1.1), applied to the canonical DAG decomposition of undirected line graphs into two directed line graphs and summing the resolvents of both DAGs (Fig. 2). Both methods are trained on the Masked Language Modeling (MLM) (Devlin et al., 2019) task on the C4 dataset (Raffel et al., 2020) for 70k steps, following the recipe used in M2 (Fu et al., 2023). The models are then fine-tuned on the GLUE benchmark. For baselines, we compare our methods with other sequence mixers such as M2 Fu et al. (2023), as well as models such as MLP-Mixer Tolstikhin et al. (2021) and FNet Lee-Thorp et al. (2022), in addition to the Transformer based BERT model Devlin et al. (2019). We choose these baselines to highlight the performance of our model when compared to different type of sequence mixing paradigms applied to large scaled language modeling tasks. We refer the reader to Appendix C for the architectural and hyper-parameter details.

From Table 1, observe that while BERT outperforms other linear baselines such as M2, MLP-Mixer, FNet it does so with an additional quadratic cost. In contrast, Chimera achieves the best of both worlds, incurring a linear time complexity while achieving strong performance. This capability arises from two key factors: first, our parameterization of the adjacency matrix allows the model to effectively modulate the influence between tokens in the sequence, leading to strong performance. Second, the structured nature of the adjacency matrix enables a fast, linear-time resolvent operation, improving the method's computational efficiency. Additionally, note that our undirected graph (UG) variant performs competitively with BERT while surpassing other recent linear baselines.

### 5.2 ImageNet-1k Classification

Table 2: Top-1, Top-5 accuracies of various methods on ImageNet-1K. Chimera outperforms the standard attention baseline ViT-B, as well as other subquadratic baselines.

| Method (88M) | Top-1 (%) | | Top-5 (%) | |
| --- | --- | --- | --- | --- |
| | Acc | Acc$_{EMA}$ | Acc | Acc$_{EMA}$ |
| ViT-B | 78.8 | 80.6 | 94.2 | 95.2 |
| S4-ViT-B | 79.4 | 80.4 | 94.2 | 95.1 |
| Hyena-ViT-B | 78.4 | 76.4 | 94.0 | 93.0 |
| Chimera-ViT-B | **81.4** | **82.1** | **95.4** | **95.9** |

Table 3: Ablation: Comparing 2D grid structure with 1D flattening of patches. We see that maintaining the 2D DAG structure outperforms method where the underlying topological structure is flattened, showing maintaining the topological structure matters.

| Method (22M) | Top-1 (%) | | Top-5 (%) | |
| --- | --- | --- | --- | --- |
| | Acc | Acc$_{EMA}$ | Acc | Acc$_{EMA}$ |
| Fwd (1D) | 73.8 | 73.8 | 91.6 | 91.6 |
| Fwd & Rev (1D) | 76.5 | 75.6 | 93.4 | 92.8 |
| 2D DAG | **77.8** | **76.7** | **93.9** | **93.5** |

We evaluate Chimera on the ImageNet-1k (Deng et al., 2009) classification task that has a grid graph topology. We compare Chimera applied to the 2D-DAG decomposition (Figure 3) topology against state-of-the-art ViT based models, a standard architecture that is used for ImageNet classification. Specifically we use ViT-B which has 88M parameters as well as other SSM based baselines like Hyena (Poli et al., 2023), S4 (Gu et al., 2022) in Table 2 to highlight how our method compares with other state-space model based architectures.

---

[3]We use a slightly modified normalization scheme for the undirected line graph method to allow for larger selectivity values in the adjacency matrix. See Appendix B.1 for details

Table 4: Evaluation of Chimera on LRGB Tasks (Dwivedi et al., 2022). The first section shows the best performing numbers cited in the papers that introduce the given baselines. The second section shows the result of better hyperparameter tuned baselines introduced by Tönshoff et al. (2023). Finally, we also compare with other baselines that use SSMs as a blackbox replacement for a Transformer. Here the best numbers are highlighted in **bold**.

| Method (< 500k params) | Peptides-Func | Peptides-Struct | PascalVOC-SP | COCO-SP |
|---|---|---|---|---|
| | AP (↑) | MAE (↓) | F1 (↑) | F1 (↑) |
| GCN (Kipf and Welling, 2016) | 0.5930±0.0023 | 0.3496±0.0013 | 0.1268±0.0060 | 0.0841±0.0010 |
| GINE (Hu et al., 2019) | 0.5498±0.0079 | 0.3547±0.0045 | 0.1265±0.0076 | 0.1339±0.0044 |
| Gated-GCN (Bresson and Laurent, 2017) | 0.5864±0.0077 | 0.3420±0.0013 | 0.2873±0.0219 | 0.2641±0.0045 |
| SAN+LapPE (Kreuzer et al., 2021) | 0.6384±0.0121 | 0.2683±0.0043 | 0.3230±0.0039 | 0.2592±0.0158 |
| Exphormer (Shirzad et al., 2023) | 0.6527±0.0043 | 0.2481±0.0007 | 0.3975±0.0037 | 0.3430±0.0108 |
| GPS+BigBird (Rampášek et al., 2022) | 0.5854±0.0079 | 0.2842±0.0130 | 0.2762±0.0069 | 0.2622±0.0008 |
| GraphGPS+Transformer (Rampášek et al., 2022) | 0.6575±0.0049 | 0.2510±0.0015 | 0.3689±0.0131 | 0.3774±0.0150 |
| GCN (Tönshoff et al., 2023) | 0.6860 ± 0.0050 | 0.2460 ± 0.0007 | 0.2078 ± 0.0031 | 0.1338 ± 0.0007 |
| Gated-GCN (Tönshoff et al., 2023) | 0.6765 ± 0.0047 | 0.2477 ± 0.0009 | 0.3880 ± 0.0040 | 0.2922 ± 0.0018 |
| GINE (Tönshoff et al., 2023) | 0.6621 ± 0.0067 | 0.2473 ± 0.0017 | 0.2718 ± 0.0054 | 0.2125 ± 0.0009 |
| GraphGPS+Transformer (Tönshoff et al., 2023) | 0.6534 ± 0.0091 | 0.2509 ± 0.0014 | 0.4440 ± 0.0054 | 0.3884 ± 0.0055 |
| Graph-Mamba (Wang et al., 2024) | 0.6739 ± 0.0087 | 0.2478 ± 0.0016 | 0.4191 ± 0.0126 | 0.3960 ± 0.0175 |
| Graph Mamba (Behrouz and Hashemi, 2024) | 0.7071 ± 0.0083 | 0.2473 ± 0.0025 | 0.4393 ± 0.0112 | 0.3974 ± 0.0101 |
| NeuralWalker (Chen et al., 2024) | **0.7096 ± 0.0078** | 0.2468 ± 0.0005 | **0.4912 ± 0.0042** | **0.4398 ± 0.0033** |
| Chimera (Ours) | 0.7021 ± 0.003 | **0.2433 ± 0.0006** | 0.4460 ± 0.007 | 0.3977 ± 0.016 |

We note that *all these baselines flatten the image into a 1D sequence and apply 1D sequence models, and do not take into account the underlying topology.* For our experiments, we simply replace the SSD layer in the Mamba block introduced in Dao and Gu (2024) with Chimera, and use the ViT-B training recipe with minimal hyperparameter tuning.

Table 2 shows that Chimera's 2D-DAG decomposition outperforms ViT by 2.6%. We note that our method does not require any additional position embeddings which are still an active area of research for ViT (Heo et al., 2024). We outperform methods such as Hyena (Poli et al., 2023) by 3%, and S4 (Gu et al., 2022) by 2% that linearize the data and then apply an SSM on it.

Furthermore, to demonstrate the importance of incorporating topology, we perform an ablation where we progressively degrade the grid-graph structure, observing a monotonic drop in performance. We consider three topologies: **2D DAG** is the 2D DAG decomposition that retains the grid structure (Fig 3, right); **Fwd & Rev (1D)** flattens the grid into a 1D sequence with bidirectional edges like ViT (Fig 5, top); **Fwd (1D)** is a 1D graph with only forward edges (Fig 5, bottom). We observe from Table 3 that as the topology is lost, the accuracy drops from 77.8% (2D-DAG) to 76.5% (Fwd & Rev) to 73.8% (Fwd).

### 5.3 Long Range Graph Benchmark

We evaluate Chimera on the Long Range Graph Benchmark (LRGB) (Dwivedi et al., 2022). This benchmark comprises tasks designed to challenge models in their ability to effectively capture both local and long-range interactions within graph structures. We compare against convolution-based (GCN Kipf and Welling (2016), GatedGCN Bresson and Laurent (2017)), Transformer-based (GraphGPS Rampášek et al. (2022)), Mamba-based (Graph-Mamba Wang et al. (2024), Graph Mamba Behrouz and Hashemi (2024)), and other

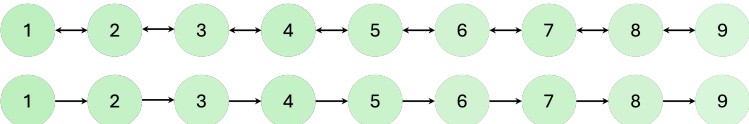

Figure 5: Progressively destroying the 2D grid graph topology. *Fwd & Rev* (top): 1D flattened grid with bidirectional edges. *Fwd* (bottom): 1D flattened grid graph with only forward edges.

baselines like GINE Hu et al. (2019), as well as their hyperparameter tuned versions introduced in Tönshoff et al. (2023). These baselines incorporate topology using a variety of techniques: convolution ones use local aggregation, transformer ones use local and global aggregation via position embeddings, and Mamba ones use "data flattening" along with random walks, position embeddings, and local encodings. The diversity of these methods highlights the significant research effort dedicated to heuristics to incorporate topology, in contrast to our unified approach.

We note that while recent methods like NeuralWalker Chen et al. (2024) achieves state-of-the-art results on LRGB, but unlike Chimera and our chosen baselines, it modifies the graph structure by sampling subgraphs via random walks and modeling them sequentially. As our goal is to evaluate models that operate directly on the original graph. That said, NeuralWalker is complementary and could potentially be combined with Chimera layers.

We show that Chimera achieves strong performance across all LRGB tasks (Table 4). Notably, we observe that on tasks such as Peptides-Func and Peptides-Struct, where convolution-based models typically outperform transformers, Chimera outperforms or matches their performance. Furthermore, on tasks like PascalVOC and COCO where transformers do well, Chimera is competitive with the best baselines. This validates our grounded approach which effectively captures both local and global information.

In Table 5, we evaluate the approximate variant of Chimera with a finite-sum relaxation (Sec 4.2) that truncates the Neumann series at the diameter of the graph. We show that the approximation variant matches the strong transformer baseline of GraphGPS, however fully leveraging the entire graph structure in Chimera provides clear performance benefits.

## 6 Discussion and Future Work

We propose Chimera, a unified framework that directly incorporates the underlying graph topology in a principled way. Unlike prior approaches that apply attention or State Space Models (SSMs) by flattening the data, we instead generalize SSMs—originally designed to operate on sequences without position embeddings—to any graph topology. We show that Chimera achieves strong performance across domains including language, vision, and graph tasks, consistently surpassing baselines, which validates our premise and the proposed solution. We further show that for the subclass of graphs which can be decomposed into DAGs, the recurrent form of Chimera affords linear complexity.

Our work has a few limitations, the most significant being that for general graphs, fully capturing all node interactions results in a cubic computational cost. This can be reduced to a quadratic cost through a straightforward approximation that truncates the infinite sum to a finite terms—determined by the diameter of the graph. That said, we still believe there is significant potential for hardware optimization just as Mamba-based methods benefited greatly from dedicated CUDA kernels. Nevertheless, we note that the current implementation of Chimera achieves a reasonable time ratio of $\sim 1.5\times$ compared to Transformer-based architectures which we believe provides a starting point for further exploration across novel domains.

We believe that developing optimized kernels for specific graph structures such as grid-graphs—along with exploring graph approximations through DAG decompositions—is a promising direction for future work. We are hopeful that the community will apply Chimera to a broader range of domains with inherent topological structures and continues to develop more efficient and performant extensions of Chimera.

Table 5: Ablation: Chimera with approximate resolvent is competitive with the Transformer baseline. This alleviates the cubic cost of evaluating the exact resolvent.

| Method | Peptides-Func | Peptides-Struct | PascalVOC-SP | COCO-SP |
| --- | --- | --- | --- | --- |
| | AP (↑) | MAE (↓) | F1 (↑) | F1 (↑) |
| GraphGPS+Transformer | $0.6534 \pm 0.0091$ | $0.2509 \pm 0.0014$ | $0.4440 \pm 0.0054$ | $0.3884 \pm 0.0055$ |
| Chimera (Approx) | $0.6979 \pm 0.0057$ | $\mathbf{0.2420 \pm 0.0013}$ | $0.4353 \pm 0.00307$ | $0.3833 \pm 0.0006$ |
| Chimera (Ours) | $\mathbf{0.7021 \pm 0.003}$ | $0.2433 \pm 0.0006$ | $\mathbf{0.446 \pm 0.007}$ | $\mathbf{0.3977 \pm 0.016}$ |

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

# A Deferred Proofs

## A.1 Proof of Proposition 3.4

*Proof.* Let $\boldsymbol{\epsilon}_i \sim \mathcal{N}(\mathbf{0}, \mathbf{I}_T)$ be $T$ i.i.d. random Gaussian vectors. Assuming Gaussian initialization for the adjacency matrix $\mathbf{A}$, it can be expressed as:

$$\mathbf{A}[i, :] = \frac{\gamma \boldsymbol{\epsilon}_i}{\|\boldsymbol{\epsilon}_i\| + \exp(-\Psi_i)}. \tag{19}$$

We first show that $\|\boldsymbol{A}\| \leq \gamma < 1$. From the concentration of the Gaussian random vector norm, $\|\boldsymbol{\epsilon}_i\| \geq \sqrt{T}/2$ for all tokens $i$, with probability $\geq 1 - \exp(T/8)$. Since $\exp(-\Psi_i) \geq 0$, $\|\boldsymbol{\epsilon}_i\| + \exp(-\Psi_i) \geq \sqrt{T}/2$. Consider any unit vector $\boldsymbol{u}$, then

$$\|\boldsymbol{A}\boldsymbol{u}\| = \sum_{i=1}^{T} \frac{\gamma \boldsymbol{\epsilon}_i^T \boldsymbol{u}}{\|\boldsymbol{\epsilon}_i\| + \exp(-\Psi_i)} \leq \gamma \sum_{i=1}^{T} \frac{2\epsilon_i}{\sqrt{T}} \leq 2\gamma \frac{\sqrt{T}\epsilon}{\sqrt{T}} = 2\gamma\epsilon < 1, \tag{20}$$

with probability greater than $1 - \Phi(\frac{-1}{2\gamma})$, were $\epsilon_i, \epsilon \sim \mathcal{N}(0, 1)$. Finally, since the operator norm of $\|\boldsymbol{A}\|$ is less than one, we apply Banach's Lemma to get,

$$\|(\boldsymbol{I} - \boldsymbol{A})^{-1}\| \leq \frac{1}{1 - \|\boldsymbol{A}\|}, \tag{21}$$

which implies that the inverse exists. $\qquad\square$

## A.2 Proof of Proposition 4.3

*Proof.*

$$\mathrm{Var}(\boldsymbol{C}_i^T \boldsymbol{h}_i) = \frac{1}{|p(i)|} \left( \sum_{j \in p(i)} \boldsymbol{A}_{ij} \mathrm{Var}(\boldsymbol{C}_i^T \boldsymbol{h}_j) + \ln(\boldsymbol{A}_{ij}) \mathrm{Var}(\boldsymbol{C}_i^T \boldsymbol{B}_i v_i) \right), \tag{22}$$

$$= \frac{1}{|p(i)|} \left( \sum_{j \in p(i)} \boldsymbol{A}_{ij} \mathrm{Var}(\boldsymbol{C}_j^T \boldsymbol{h}_j) + \frac{2}{d} \ln(\boldsymbol{A}_{ij}) \right), \tag{23}$$

where we have used the fact that $\mathrm{Var}(\boldsymbol{C}_j^T \boldsymbol{h}_j) = \mathrm{Var}(\boldsymbol{C}_i^T \boldsymbol{h}_j)$, and that the variance of $\mathcal{X}^2$ distribution with $d$ degrees of freedom is $2d$. Let $d \geq 4$, then

$$\mathrm{Var}(\boldsymbol{C}_i^T \boldsymbol{h}_i) \leq \frac{1}{|p(i)|} \left( \sum_{j \in p(i)} \boldsymbol{A}_{ij} + \frac{2}{d} \ln(\boldsymbol{A}_{ij}) \right) \leq \frac{1}{|p(i)|} \sum_{j \in p(i)} 1 \leq 1, \tag{24}$$

where we have used the fact that $\boldsymbol{A}_{ij} \in [0, 1]$. $\qquad\square$

## A.3 Proof of Proposition 4.4

*Proof.* In the structured masked attention (SMA) framework Dao and Gu (2024), the computational complexity is the cost of the matrix-vector multiplication by the mask matrix $\boldsymbol{L} = (\boldsymbol{I} - \boldsymbol{A})^{-1}$. For DAGs, $\boldsymbol{A}$ is (up to conjugation by a permutation) a *lower-triangular* matrix with $|\mathcal{E}|$ nonzero entries. Computing $\boldsymbol{y} = (\boldsymbol{I} - \boldsymbol{A})^{-1} \boldsymbol{x}$ reduces to solving the system $(\boldsymbol{I} - \boldsymbol{A})\boldsymbol{y} = \boldsymbol{x}$ via forward substitution.

We perform Gaussian elimination by iterating over the ordered list $\{0, \ldots, |\mathcal{V}| - 1\}$ and choosing the pivots $(i, i)$. Since $\boldsymbol{I} - \boldsymbol{A}$ is lower-triangular, each pivot operation affects only a single column rather than the entire row, reducing the cost per step to $O(\mathrm{nnz}(\mathbf{A}[:, i]))$, where $\mathrm{nnz}(\cdot)$ denotes the number of non-zero entries. Summing over columns, the complexity is,

$$O\left( \sum_{i}^{|\mathcal{V}|} \mathrm{nnz}(\mathbf{A}[:, i]) \right) = O(\mathrm{nnz}(\mathbf{A})) = O(|\mathcal{V}| + |\mathcal{E}|).$$

For our motivating example of Mamba, $\boldsymbol{I} - \boldsymbol{A}$ has exactly $2|\mathcal{V}|$ nonzero entries, ensuring a linear-time complexity. $\qquad\square$

### A.4 Proof of Theorem 4.5

*Proof.* **Backward pass.** The local update rule of backpropagation requires applying the chain rule through the matrix inverse operation, in particular, using the following identity applied to $\boldsymbol{Y} = (\boldsymbol{I} - \boldsymbol{A})$,

$$\frac{\partial \boldsymbol{Y}^{-1}}{\partial \theta} = -\boldsymbol{Y}^{-1} \frac{\partial \boldsymbol{Y}}{\partial \theta} \boldsymbol{Y}^{-1} \tag{25}$$

Because $\boldsymbol{Y}^{-1}$ is already computed in the forward pass, it can be cached, and then the marginal cost of the local backpropagation is simply two extra matrix multiplications.

**Forward pass.** To compute $\boldsymbol{L} = (\boldsymbol{I} - \boldsymbol{A})^{-1}$ more efficiently for DAGs, we leverage the equivalence of Neumann series to the series $\boldsymbol{L} = \boldsymbol{I} + \boldsymbol{A} + \boldsymbol{A}^2 + \cdots$, which comes to a finite sum for DAGs due to the nilpotence of $\boldsymbol{A}$ matrix. We compute this sum more efficiently using the "squaring trick" as,

$$(\boldsymbol{I} - \boldsymbol{A})^{-1} = (\boldsymbol{I} + \boldsymbol{A})(\boldsymbol{I} + \boldsymbol{A}^2)(\boldsymbol{I} + \boldsymbol{A}^4) \cdots (\boldsymbol{I} + \boldsymbol{A}^k), \tag{26}$$

where $k$ is the smallest power of 2 larger than the graph diameter $\mathrm{dia}(\mathcal{G})$. This can be computed using $O(\log(\mathrm{dia}(\mathcal{G})))$ matrix multiplications to compute the powers of $\boldsymbol{A}$ for powers-of-two exponents, and then $O(\log(\mathrm{dia}(\mathcal{G})))$ matrix multiplications to multiply together the right-hand side. $\qquad\square$

# B  Additional Experiments

## B.1  MLM: Chimera on Undirected Line Graphs

For an undirected line graph (Figure 2, left), the adjacency matrix $\boldsymbol{A}$ takes the following form:

$$\boldsymbol{A} = \begin{bmatrix} 0 & a_{12} & 0 & \cdots & 0 \\ a_{21} & 0 & a_{23} & \cdots & 0 \\ 0 & a_{32} & 0 & \cdots & 0 \\ \vdots & \vdots & \vdots & \ddots & \vdots \\ 0\cdots 0 & 0\cdots 0 & 0 & a_{T-1,T} & 0 \end{bmatrix}.$$

As discussed in Section 3.3, to ensure the existence of $(\boldsymbol{I} - \boldsymbol{A})^{-1}$, we introduced a row-wise sum normalization strategy, wherein we normalized each row of the adjacency matrix with $\sum_j \boldsymbol{A}_{ij} + \Psi_i$. However, since this constraint is designed for general graphs, it is not sufficiently expressive. Therefore, we instead use a strictly more expressive constraint for line graphs which enforces $\boldsymbol{A}_{ij} \cdot \boldsymbol{A}_{ji} + \Psi_i \leq \frac{1}{4}$ on each simple cycle of the graph.

**Proposition B.1.** *Under the above constraint, the inverse $(\boldsymbol{I} - \boldsymbol{A})^{-1}$ exists as for any two nodes, the sum of all paths between them is upper bounded by $\sum_i (1/4)^i \leq 1/3$.*

## B.2  Imagenet: Parameter Sharing Ablation

We study the trade-off between sharing parameters for $\boldsymbol{B}, \boldsymbol{C}$ across different graphs as a domain-dependent design choice. We explore four settings: *No sharing, Complete sharing, Row-wise sharing, and Diagonal sharing* across the four DAGs. From Table 6, we observe that diagonal sharing achieves the best performance, indicating it strikes the optimal tradeoff between parameter sharing and other modes of increasing expressivity for modeling image data.

| Method (22M) | Top-1 (%) | | Top-5 (%) | |
|---|---|---|---|---|
| | Acc | Acc$_{\text{EMA}}$ | Acc | Acc$_{\text{EMA}}$ |
| None | 77.10 | 76.13 | 93.55 | 93.15 |
| Complete | 77.25 | 76.09 | 93.75 | 93.21 |
| Row-wise | 77.46 | 76.57 | 93.76 | 93.37 |
| Diagonal | **77.80** | **76.69** | **93.87** | **93.53** |

Table 6: Ablation: Diagonal parameter sharing works best.

## B.3  Chimera When Both # Layers and # Parameters Are Controlled

Chimera's architecture builds on the Mamba Block from Mamba-2, which utilizes a greater number of layers than Transformers due to its higher parameter efficiency. In this section, we conduct an ablation study where both the number of layers and total parameter count are controlled by adjusting the expansion factor $e$ in the Mamba Block to 4. Specifically, we compare three models on the bidirectional language modeling task:

- **Chimera-12L**: 12 layers, baseline configuration with 70M parameters.

- **BERT-6L**: 6-layer Transformer baseline with 70M parameters.

- **Chimera-6L**: 6 layers with an expansion factor of 4, maintaining the same parameter count as the other models.

To reduce computational costs, we train these models on a reduced ablation setting with 98M steps instead of the standard 245M steps and the results are summarized in the table below. Notably, Chimera-6L and Chimera-12L achieve nearly identical performance, both significantly outperforming BERT-6L. This

Table 7: Ablation: Chimera maintains strong performance when both number of layers and number of parameters are controlled.

| Model | Evaluation Metrics | |
| --- | --- | --- |
| | Masked Accuracy (↑) | Cross-Entropy Loss (↓) |
| BERT-6L | 0.6176 | 1.9466 |
| Chimera-6L | **0.6360** | **1.8108** |
| Chimera-12L | **0.6363** | **1.8142** |

demonstrates that Chimera's improvements are not simply a result of increased depth but rather stem from its core methodological advancements.

## C    Architectural Details

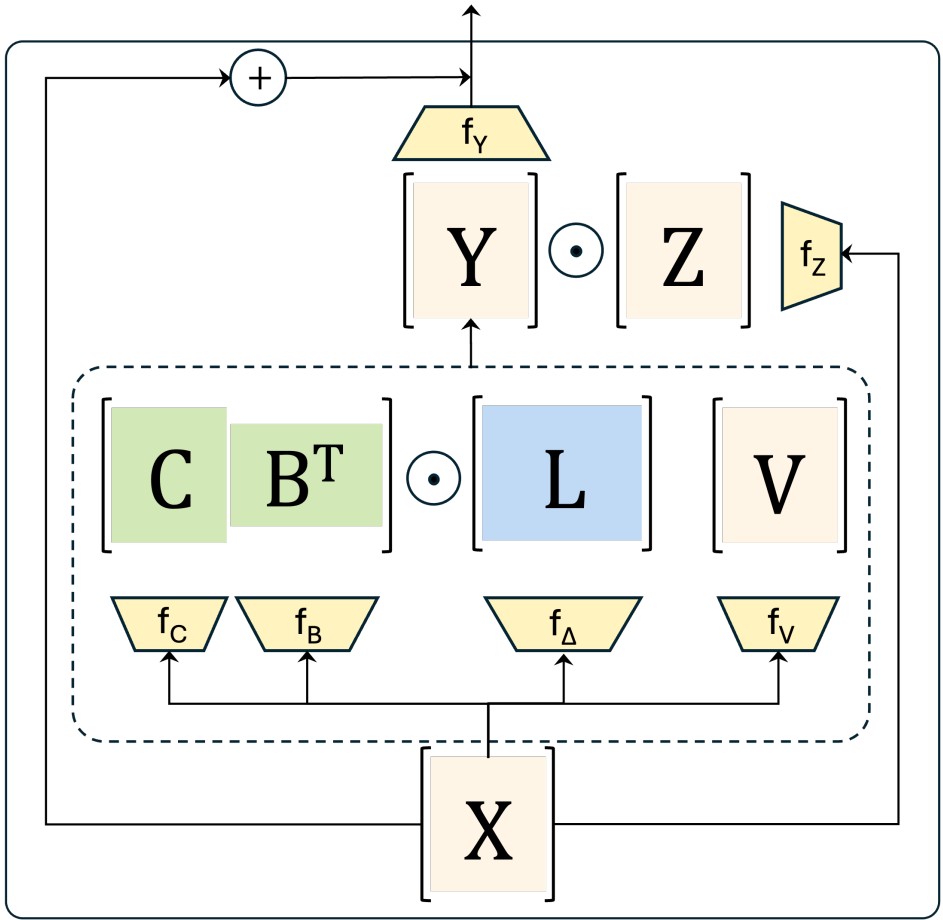

Figure 6: Chimera's Architecture: The output of the Chimera layer is embedded within the gated block introduced in Mamba-2 (Dao and Gu, 2024). Here $X$ matrix denotes the input to the block, and $f_c, f_B, f_\Delta$ and $f_V$ are data dependent projections defined in Section 2. The operator $\odot$ denotes element-wise multiplications between matrices, and $\oplus$ defines addition. The output from the Chimera layer is passed through a Gated-MLP, a final projection $f_Y$, followed by a residual connection.

### C.1    Masked Language Modeling

In Table 8, we provide the architectural and training details for BERT-B and Chimera on the MLM task. For both the models, we follow the M2 recipe from Fu et al. (2023), adjusting the number of layers to 12 for BERT-B and 23 for Chimera to control for the number of parameters. We conducted a small sweep to fine-tune the learning rate for Chimera, choosing $8e - 4$ over BERT-B's $5e - 4$.

### C.2    Imagenet-1k Classification

For the image classification experiments, we largely follow the ViT-B recipe with the following adjustments as shown in Table 9, where the hyperparameters are carefully tuned—and hence different from Table 8—in order to perform best on the image classification task. To control for the number of parameters, we adjust the number of layers from 12 for ViT-B to 22 for Chimera. Additionally, we reduce the Cutmix augmentation from 1.0 to 0.1, as Chimera's stronger inductive bias mitigates the risk of overfitting.

Table 8: Architectural and Training Details for BERT-B and Chimera on MLM

| Parameter | BERT-B (110M) | Chimera (110M) |
|---|---|---|
| Model dimension ($d_{\text{model}}$) | 768 | 768 |
| Layers | 12 | 23 |
| Max sequence length | 128 | 128 |
| Num Heads | 12 | 12 |
| Head size | 64 | 64 |
| Optimizer | Decoupled AdamW | Decoupled AdamW |
| Learning rate | $5e-4$ | $8e-4$ |
| Optimizer momentum | $\beta_1 = 0.9, \beta_2 = 0.98$ | $\beta_1 = 0.9, \beta_2 = 0.98$ |
| Weight decay | $1e-5$ | $1e-5$ |
| Batch size | 4096 | 4096 |
| Learning rate schedule | Linear decay with warmup | Linear decay with warmup |
| Training steps | 70k | 70k |
| MLM Probability | 0.3 | 0.3 |

In Table 10, we present the reduced setting used for our ablation studies in Tables 6 and 3, where we match the number of parameters of ViT-S (22M).

Table 9: Hyperparameters used for ViT-B and Chimera for ImageNet-1k classification task

| Parameter | ViT-B (88M) | Chimera (88M) |
|---|---|---|
| Image size | $224^2$ | $224^2$ |
| Optimizer | AdamW | AdamW |
| Optimizer momentum | $\beta_1, \beta_2 = 0.9, 0.999$ | $\beta_1, \beta_2 = 0.9, 0.999$ |
| Weight init | trunc. normal (std=0.02) | trunc. normal (std=0.02) |
| Learning rate | $1e-3$ | $1e-3$ |
| Weight decay | 0.05 | 0.05 |
| Batch size | 1024 | 1024 |
| Training epochs | 310 | 310 |
| Learning rate schedule | cosine decay | cosine decay |
| Warmup epochs | 10 | 10 |
| Warmup schedule | linear | linear |
| Patch Size | 16 | 16 |
| Layers | 12 | 22 |
| Num Heads | 12 | 12 |
| Droppath | 0.3 | 0.3 |
| Randaugment | (9,0.5,layers=2) | (9,0.5,layers=2) |
| Mixup | 0.8 | 0.8 |
| Cutmix | 1.0 | 0.1 |
| Random erasing | 0.25 | 0.25 |
| Label smoothing | 0.1 | 0.25 |
| Stochastic depth | 0.1 | 0.25 |
| Exp. mov. avg (EMA) | 0.99996 | 0.99996 |

## C.3 Long Range Graph Benchmark

To train Chimera on the Long Range Graph Benchmark we follow a similar training recipe to that provided in Rampášek et al. (2022) where we replace the Transformer layers with Chimera layers. Moreover, in line with the baselines, we make sure that our models have less than $500k$ parameters. While training Chimera on

Table 10: Key differences between the original and the ablation setting for Chimera

| Parameter | Chimera-S (2D) |
|---|---|
| Model dimension ($d_{\mathrm{model}}$) | 384 |
| Number of layers | 22 |
| Number of Heads | 3 |
| Droppath | 0.1 |

graphs we remove the Gated-MLP layer $Z$ defined in Figure 6. We did this to keep our training recipe as close to that provided in Rampášek et al. (2022) and highlight the effectiveness of Chimera. The hyperparameters used to train Chimera are provided in Table 11.

Table 11: Hyperparameters running Chimera on the Long Range Graph Benchmark

| | Peptides-Func | Peptides-Struct | PascalVOC-SP | COCO-SP |
|---|---|---|---|---|
| Learning Rate | 0.001 | 0.0015 | 0.0035 | 0.0035 |
| Optimizer | Adam | Adam | Adam | Adam |
| dropout | 0.1 | 0.1 | 0.05 | 0.05 |
| #layers | 8 | 8 | 8 | 8 |
| hidden dim. | 64 | 64 | 64 | 64 |
| hidden state dim. | 96 | 80 | 64 | 64 |
| num heads | 2 | 4 | 4 | 4 |
| batch size | 64 | 64 | 32 | 32 |
| #epochs | 200 | 200 | 200 | 200 |
| norm | LayerNorm | LayerNorm | LayerNorm | LayerNorm |
| MPNN | GCN | GCN | GCN | GCN |
| #Param. | 499k | 504k | 489k | 489k |

