# OpenReview forum: "Chimera: State Space Models Beyond Sequences"
_TMLR — Accepted by TMLR_

### Review · Reviewer_VABZ · 2025-06-23

**Summary Of Contributions:**

This work proposes Chimera, a general-purpose model that generalizes state space models. The authors propose to incorporate data topology (more specifically, graph topology) into the update computation of hidden states in a principled way. Additionally computational efficiency of Chimera is proven. Chimera achieves state-of-the-art performance in at least some vision, language, and graph tasks.

**Audience:**

Yes

**Broader Impact Concerns:**

I have no such concerns.

**Claims And Evidence:**

No

**Requested Changes:**

- Address and fix the long list of missing definitions, unclear or incorrect statements, and missing background and discussion. (critical due to their number)
- Provide your code so we the reviewers can check the correctness of your claims (critical)
- Adjust your figure and table placement (strengthen)
- Make sure to make each figure and table self-contained, and make sure each figure and table is referenced appropriately (namely, Figure 2). (critical)
- Deal with or respond to my issue with A.1, the proof of Proposition 3.4. (critical)
- Address your choice of baselines (critical)
- Address your choice of hyperparameters (critical because the code is not available, otherwise it might only strengthen)
- Provide standard deviations (critical if possible)
- Use standard deviations to highlight significant differences. A standard way to do this would be to underline those values that are not the best, but that are only insignificantly worse than the best value, sharing first place or boldness.

**Strengths And Weaknesses:**

# Strengths:
- The general approach taken is interesting and seems principled.
- The experiments are compelling.
- Most statements and approaches are straightforward.
- Most statements are proven to be theoretically robust.

# Weaknesses:
- As far as I can see, you do not provide code while claiming state-of-the-art performances on 3 significant benchmarks. I doubt the veracity of these claims, but I cannot check their correctness. ()
- While the used language and organization are well done, there are multiple issues with missing definitions, unclear or incorrect statements, or missing background and discussion.
  - In the Introduction, you state, "Typically, Transformer-based methods [...] are used to model [molecule data]." This statement is at the very least questionable. While Transformer-based GNN approaches have become more prevalent, simple message-passing approaches, which respect graph topology naturally, are still very common.
  - In the Introduction, you state, "while absolute position embeddings have inherently constrained context size." This statement is incorrect, or at the very least requires a non-standard definition of absolute position embeddings. For instance, the random-walk structural encoding is not context-constrained for long enough walks. The resistance distance positional encoding is not context-constrained at all. Arguably, the Laplacian Positional Embedding is also not context-constrained.
  - In Equation 4 "\Delta_t" is not defined.
  - In Proposition 3.2, "weighted adjacency matrix" is not defined. Specifically, the term weighted in this context is not defined.
  - In Proposition 3.3, "w_{i-1 \to i}" is not defined and is an unusual notation in GNN contexts.
  - In between Equations 11 and 12, "Z" is not defined.
  - In Equation 12, "\Delta' " is not defined.
  - After Equation 12 "\Delta^{(1)}" and "\Delta^{(2)}" are not defined or clearly described.
  - Multiple functions like "f_{\Delta'}" and "f_{\Psi}" are said to be similarly computed to other functions, but this is not defined.
  - The statement that line graphs and grid graphs can be canonically decomposed into multiple DAGs is not conceptualized. What does this mean in your context?
  - It is not defined what the parents of a node in a DAG are exactly. There are two possibilities due to directedness.
  - In Section 4.1.1, "\Delta_i[j]" is not defined
  - In Proposition 4.1, "A^T" is not a good choice, namely, the T resembles the transpose of matrix A, but you are referring here to the T-th power.
  - Repeatedly, you use channels to describe the number of features, but you never define what channels are.
  - In Proposition 4.2, "\mathbf{v}" does not show up in Equation 14 unless you mean "v_i" to be the i-th element of "\mathbf v"
  - In Proposition 4.3, you assume that "B_iv_i, C_i \sym \mathcal N(0,I_d)." This assumption is not justified.
  - In all tables, boldness and underlining are not defined.
  - In Section 4.2, you refer to the adjacency matrix of a graph topology defined in Section 3.3. However, it is generally unclear when A is referring to the typical adjacency matrix and when A is referring to your modified version of Equation 12 or the Equation after Equation 12, or both.
  - In Section 5.1, you propose for the first time to use the canonical DAG representation of line graphs by summing the resolvents. This should be made clear explicitly, how this fits into your equations.
- Almost all figures and tables are oddly positioned. Figure 1 on page 4 is only referenced on page 5. Figure 2 on page 7 is never referenced. Figure 3 and Figure 4 on pages 8 and 9 are referenced on pages 2, 6, 9, 10, and 11. Table 1 on page 8 is only referenced on page 10. Similarly, all other tables are misplaced, except for Table 3, which is placed in a section mid-text, which I would advise against.
- All tables and figures are not self-contained.
- In A.1 Proof of Proposition 3.4, you state "||\epsilon_i||\geq \sqrt{T}" whp. This is incorrect. The norm of a normal Gaussian vector is concentrated around \sqrt{T} with its mean slightly below \sqrt{T}. This is a significant statement required for the computation of Chimera in general and needs to be resolved. You also state that "\epsilon_i \sym \mathcal T (0, I_T)." Why is the distribution labelled as "\mathcal T" instead of "\mathcal N"?
- You do not discuss why they chose the baselines in each of their experiments, or why these baselines are appropriate, and do not discuss whether the values used in their tables are copied from previous work or computed anew.
- Similarly, you make hyperparameter changes in tables 8 and 9, but do not justify them.
- You do not report standard deviations in most of your tables; however, these are important to assess the significance of the difference. Even when the std is not available for other methods (because you perhaps copied them from another paper), they could justify whether your achieved performances are significantly better than previously reported averages.
- In extension to the previous statement, you do not make use of standard deviations to asses significance of difference. For instance, in Table 2, PascalVOC-SP Chimera does not significantly outperform GraphGPS. On COCO-SP, it does not significantly outperform either Graph Mamba.

---

> ### Author Response · Authors · 2025-07-25
>
> We thank the reviewer for their detailed feedback! We have tried to address most of the comments and the edits that the reviewer has mentioned in our submission, with the changes are highlighted in blue! We are also glad that the reviewer finds our results and experiments to be robust and compelling!
>
> We would now like to address some of the comments by the reviewer below:
>
> **On Reporting Standard Deviations**
>
> We appreciate the reviewer’s emphasis on statistical rigor. For the graph classification benchmarks, we do report standard deviations across multiple runs. However, for the language and image-based graph tasks, we followed common practice in prior work and reported single-run results, as each run is computationally intensive—often requiring multiple GPUs and several days to complete. Moreover, we have observed that Chimera’s performance on these tasks is *consistently stable across runs*, with minimal variance, making single-run reporting a reasonable approximation. We will add a sentence clarifying this in the final draft.
>
> **Significance of difference on LRGB with other baselines**
>
> We agree with the reviewer that considering the standard deviation for the graph benchmarks Chimera only marginally outperforms other baselines. However, our primary goal in this work is to demonstrate that Chimera provides a *unified and effective framework* for graph learning across diverse modalities—including language, vision, and general graphs—without relying on modality-specific positional encodings or architectural adjustments. We show that a consistent and straightforward application of Chimera layers achieves competitive, and in many cases, state-of-the-art performance across benchmarks. This broad applicability highlights Chimera’s strength as a *versatile and robust approach* to graph representation learning.
>
> **On the Canonical Decomposition of Graphs**
>
> By “canonical” we simply mean the *obvious*, structure‑induced orientation of edges that yields DAGs in these simple cases:
>
> - **Line graphs:** orient edges $1\to2\to\cdots\to V$.
> - **2D grids:** perform four natural grid scans from any of the four vertices.
>
> We agree with the reviewer and recognize that a single “canonical” choice may not exist for arbitrary topologies.
>
> **On the parents of nodes in a DAG**
>
> In this work we define the root as the node with only outgoing edges, and parents of a node j are all nodes i, such that (i,j) is a directed edge.

---

### Review · Reviewer_yENo · 2025-07-02

**Summary Of Contributions:**

This paper introduces Chimera, a unified model that generalizes SSMs to directly incorporate graph topology in a principled manner. To enhance efficiency, the authors propose algorithmic optimizations by specializing Chimera to DAG and approximating the resolvent with a finite sum, which works well for data modalities like images and language. Experiments show that Chimera achieves strong and consistent performance across various domains, outperforming BERT on GLUE, ViT on ImageNet-1k, and competitive baselines on the Long Range Graph Benchmark, effectively modeling both local and global interactions while respecting graph structure.

**Audience:**

Yes

**Broader Impact Concerns:**

No concerns

**Claims And Evidence:**

Yes

**Requested Changes:**

**Major Comments**

I suggest that the manuscript should be improved along several directions.

First, it is somewhat difficult to figure out the time and space complexities of Chimera for computing the resolvent matrix in both the language and image domains. As I understand it, the time complexity can be reduced to linear, while the memory requirement remains quadratic in the sequence length. It would be helpful to summarize the time and space complexities of the network’s forward pass in a table, as this would effectively highlight its lower computational burden compared to Transformers.

Additionally, in Section 4.1.2, it was unclear how the proposed parallelized computation of the resolvent is implemented in linear time. I had difficulty following the explanation, and it would benefit from a more detailed or illustrative description to clarify how this is implemented.

For the general topology, it appears that both time and space complexities remain quadratic with respect to sequence length. If this is the case, could the authors clarify the advantages of Chimera over Transformers in such settings? I was unable to find empirical evidence supporting this claim.

**Minor Comments**
Figure 2 contains unnecessary lines, likely artifacts from a screenshot. It would be more professional to include a clean vector graphic.
The placement of Figures 3 and 4 does not align well with their descriptions in the text. Consider moving them to page 7 for better readability.

**Strengths And Weaknesses:**

**Strengths**

The major strength of this work lies in its principled approach to eliminating heuristics for incorporating data topology within the SSM framework. It is particularly noteworthy that the authors achieve better performance than existing SSMs, despite those models already encoding some knowledge of sequence topology. This result highlights the effectiveness of directly incorporating topology as a powerful inductive bias across diverse domains such as language, images, and graphs, without relying on domain-specific heuristics.

**Weaknesses**

My primary concern with this paper is the lack of clarity in the manuscript, which would benefit from careful revision and final polishing. In particular, the discussion of time and space complexities, especially in comparison to Transformers and SSMs under general topologies, is unclear. As I understand it, for general sequence topologies, the model does not offer any complexity reduction and remains quadratic with respect to sequence length. Please refer to the requested changes section for more detailed suggestions.

---

> ### Author Response · Authors · 2025-07-25
>
> We are glad that the reviewer found our novel approach of directly incorporating data topology into the model both *principled* and *effective*. The reviewer also found it noteworthy that Chimera is better performant than other models that already encode knowledge of sequence topology .
>
> We now take this opportunity to respond to their questions and concerns.
>
> ### On Chimera’s computational and memory complexity
>
> We begin with a quick summary (and their references in our paper) of the computational cost of Chimera’s different variants.
>
> 1. On General Graphs
>     1. **Exact Resolvent Computation (Detailed in Section 3.3; Used in Table 4)**
>
>     The method incurs a cubic $O(V^3)$ compute cost and a quadratic $O(V^2)$  memory cost, where $O(V^2)$ is the number of nodes of the graph, due to the inversion operation
>
>
>     2. **Approximate Resolvent Computation (Detailed in Section 4.2; Used in Table 5)**
>
>     Using a finite sum **approximation** of the resolvent, we reduce the compute cost to quadratic $O(V^2)$ , while the memory remains quadratic $O(V^2)$.
>
>
> 2. On General Directed Acyclic Graphs (DAGs)
>     1. **Recurrent Resolvent Computation (Detailed in Section 4.1.2, Proposition 4.4)**
>
>     Method has a linear cost in compute and memory $O(V+E)$ where $V,E$  are the number of edges and vertices respectively. One **cannot** improve this cost as simply reading the graph incurs a linear cost. However, this is a sequential operation that is not amenable to modern hardware accelerators like GPUs.
>
>     2. **Parallel Resolvent Computation (Detailed in Section 4.1.2, Theorem 4.5 and its proof)**
>
>     The method incurs a quadratic cost in compute and memory, $O(V^2)$.  *(Reference: “Although this technique is highly parallelizable, it requires the materialization of the adjacency matrix which is quadratic in the number of nodes, $V$.”)* Consider the worst case: DAG where $(i,j)$ is an edge if $i$ appears before $j$ in the topological order. Here, the number of edges are of $E = O(V^2)$ and by the $O(V+E)$ lower bound, the cost is optimal.
>
> 3.  **On Special Directed Acyclic Graphs**
>     1. **Line Graphs for Language (Used in Table 1)**
>
>     Method has a linear cost in both the recurrent mode (*Section 4.1.2) and parallel mode since we can directly use Mamba kernels.*
>
>     2. **Grid Graphs for Images (Used in Table 2, 3)**
>
>     Method has a linear cost in both the recurrent mode (*Section 4.1.2),* however, we use the general DAG kernel in the parallel mode with a quadratic cost *(reference: “… resolvent efficiently on modern hardware accelerators by leveraging matrix multiplications, although at the cost of quadratic total FLOPs.”)*
>
>     We have noted in the limitations that writing a linear kernel for domain specific DAGs, like grid graphs, is an important future direction of work. Observe that since $E = O(V)$, the pessimistic lower bound does not apply here. However, at this point, we consider it to be out of scope. Instead, we have position our work as:
>
>     1) A methodological contribution that does NOT require bespoke position embeddings as the inductive bias.
>
>     2) Providing empirical evidence that the method is competitive with state-of-the art methods across diverse domains.
>
>     3) Providing a general and a reasonably fast kernel for quick experimentation by the domain specific community. We sincerely hope that these communities will contribute towards this line of work with faster domain-specific kernels.
>
>     We have now modified the introduction, methods, and conclusion section to better clarify the computational and memory requirements of Chimera.
>
>
> ### Advantages over Transformers
>
> 1. **Unified topology inductive bias with no positional encoding engineering**
>
>     In this work we have demonstrated that in domains such as, text, images, graphs, Chimera matches or exceeds their accuracy of Transformer baselines that carefully design and learning positional/structural embeddings. However, **only Chimera** naturally extends out‑of‑the‑box to new upcoming domains—genomics (branching DNA/RNA, drug discovery), hierarchical or relational data—where no bespoke embeddings yet exist or have been standardized.
>
> 2. **Scope to improve compute on structured graphs**
>
>     Our general DAG implementation incurs $O(V^2)$ time and memory in the worst case. However, many real‑world topologies (line graphs, trees, grids) admit truly linear or sub‑quadratic algorithms (Sec 4.1.2). Developing fused CUDA/Triton kernels for these common structures is a future path to restoring linear‑time performance in practice.

---

> > ### Comment · Reviewer_yENo · 2025-08-14
> >
> > Thank you for providing detailed clarifications in the rebuttal. Your explanation of the computational and memory complexities across the different Chimera variants, covering general graphs, DAGs, and special cases like line and grid graphs, has helped me understand the trade-offs much more clearly. I particularly appreciate the explicit references to sections, propositions, and the complexity bounds, which make the reasoning transparent.
> >
> > I remain positive about the versatility of this approach. The principled integration of topology into the SSM framework, without relying on domain-specific positional embeddings, is a notable strength, especially given its applicability across diverse modalities such as language, vision, and graphs. The discussion about how Chimera can naturally extend to new domains such as genomics and hierarchical data also reinforces its potential long-term impact.

---

### Review · Reviewer_z8NQ · 2025-07-13

**Summary Of Contributions:**

In this work, the authors develop a generalization of state-space models (SSMs) to directly leverage the topology of the processed data, which could be sequences, pixel grids, or general graphs. Compared to the common approach of applying transformers to such data and adding appropriate positional embeddings, the authors instead design Chimera, a SSM which can process the data in a principled way. Further, the authors suggest an efficient implementation of Chimera, decomposing graph topologies into directed acyclic graphs (DAGs), a graph class for which efficient, and GPU-parallelized implementations exist. For example, line and grid graphs, such as those appearing in language modeling and vision, can be decomposed into a combination of DAGs. For general graphs, the authors propose an approximation technique which allows Chimera to run in quadratic time. A series of experiments reveals that Chimera can outperform classical deep learning models for language and vision, such as BERT or ViT. Further, on general graph tasks, Chimera is competitive with state-of-the-art graph models.

**Audience:**

Yes

**Broader Impact Concerns:**

None.

**Claims And Evidence:**

Yes

**Requested Changes:**

The following suggestions are derived from *Weaknesses* and are *not* critical to my assessment of the paper:

* Adding a discussion on challenges with efficient CUDA implementations of Chimera. While the authors already list the lack of specialized implementations as limitations of Chimera, it would be highly interesting what challenges the authors faced or foresee in implementing Chimera efficiently.
* Updating the LRGB table to include stronger baselines, e.g., NeuralWalker.

**Strengths And Weaknesses:**

### Strengths

* The observation that many state-of-the-art deep learning models depend on heuristic choices of positional embeddings is crucial. As such, architectural alternatives which model data in a more principled way are a highly important and timely research direction.
* The paper is written very clearly and is well motivated.
* I especially appreciate that the authors consider language, vision, and general graph tasks, instead of just focussing on one. As a result, the authors convinced me of the generality of Chimera.
* The focus on implementation efficiency, especially for modern GPU hardware, is important and greatly strengthens the contributions of the paper.

### Weaknesses
* Instead of focussing on an approximation method to reduce the complexity of Chimera on general graphs, the authors should consider efficient CUDA implementations to speed up computation, or discuss potential difficulties such an implementation.
* The fact that, on general graphs, the approximative Chimera still requires quadratic complexity might be a limiting factor when applying Chimera to general problems in graph learning (which often require linear-time models to be feasible).
* The selection of the baselines for the LRGB benchmark is not fully representative of the current state-of-the-art. For example, NeuralWalker (https://arxiv.org/abs/2406.03386v2) beats Chimera on 3/4 datasets and clearly outperforms it on 2 (COCO and Pascal).

---

> ### Author Response · Authors · 2025-07-25
>
> Thank you for your review, we are glad that the reviewer found our paper promising and our results on all the three modalities, i.e, language, images and graph convinced them of the generality of Chimera!
>
> We address some the concerns by the reviewer below:
>
> **On more efficient CUDA implementation and its potential difficulties**
>
> Our primary goal in this work was to introduce a methodological shift for modeling data with an underlying graph topology, with a lesser emphasis on implementation optimizations. While Chimera’s current naive PyTorch implementation is slow, there is substantial potential to reduce wall clock time through hardware-aware kernels. Developing such kernels, however, is a non-trivial task. For instance, the first version of Mamba [1], which used a parallel scan algorithm for sequence mixing, was 15x slower than FlashAttention-2 [3, 4] (state size=256, sequence length<1000) and required over 2,400 lines of CUDA code. Mamba achieved parity with FlashAttention-2 only in its second iteration, Mamba-2 (SSD) [2], which involved over 6,400 lines of Triton code developed by the creators of FlashAttention. Given that Chimera is a more involved method compared to Mamba-2, we view the development of a fast kernel as an exciting opportunity for future work.
>
> **On Quadratic Cost for General Graphs**
>
> We fully agree with the reviewer that, for an arbitrary graph—absent any exploitable structure—the quadratic complexity is a fundamental lower bound in our framework for faithfully modeling its topology; there is no free lunch. Indeed, this cost arises directly from the need to materialize the $V \times V$ adjacency matrix.
>
> However, when the matrix is sparse—i.e. $\text{nnz}(A)=o(V^2)$—one can avoid full materialization and instead leverage CUDA’s sparse‐matrix libraries (e.g. cuSPARSE) to reduce both memory and computation.
>
> **On adding more performant Neural Walker as a baseline**
>
> We thank the reviewer for suggesting a comparison with NeuralWalker. It is indeed a strong baseline for graph datasets, achieving state-of-the-art performance on several LRGB benchmarks, including COCO and Pascal.
>
> However, our focus with Chimera is fundamentally different: we aim to develop a *simple, unified architecture* that generalizes across diverse modalities—images, language, and graphs—*without* relying on domain-specific design choices or altering the underlying structure. NeuralWalker introduces structural modifications by sampling subgraphs via random walks and modeling them sequentially, effectively transforming the original graph. In contrast, Chimera operates directly on the native structure, maintaining consistency across modalities. We view NeuralWalker as complementary to our work—its walk-based sampling could potentially be integrated with Chimera layers for further exploration.
>
> We have now included a discussion of NeuralWalker in our revised manuscript.
>
> [1]: Albert Gu, Karan Goel, and Christopher Ré. Efficiently modeling long sequences with structured state spaces. ICLR, 2022.
>
> [2]: Tri Dao and Albert Gu. Transformers are SSMs: Generalized models and efficient algorithms through structured state space duality. In International Conference on Machine Learning (ICML), 2024b.
>
> [3]: Tri Dao and Albert Gu. State Space Duality (Mamba-2) Blogpost [Link](https://goombalab.github.io/blog/2024/mamba2-part1-model/#the-ssd-model)
>
> [4]: Tri Dao. FlashAttention-2: Faster Attention with Better Parallelism and Work Partitioning

---

### Decision · Action_Editor_rVyN · 2025-08-29

**Recommendation:** Accept as is

**Audience:**

Yes

**Audience Explanation:**

The proposed modality-agnostic architecture that incorporates underlying graph structure in a principled way represents an architectural improvement, which is a very interesting and important topic in the deep learning community.

**Claims And Evidence:**

Yes

**Claims Explanation:**

As all the reviewers noted, the claims made in this paper are well supported by several theoretical justifications and experimental results.